# Ergonomics Design and Assistance Strategy of A-Suit

**DOI:** 10.3390/mi13071114

**Published:** 2022-07-15

**Authors:** Leiyu Zhang, Xiang Gao, Ying Cui, Jianfeng Li, Ruidong Ge, Zhenxing Jiao, Feiran Zhang

**Affiliations:** 1Beijing Key Laboratory of Advanced Manufacturing Technology, Beijing University of Technology, Beijing 100124, China; zhangleiyu@bjut.edu.cn (L.Z.); gaoxiang97@foxmail.com (X.G.); jiaozhenxing0109@163.com (Z.J.); 2China-Janpan Friendship Hospital, Beijing 100029, China; 15011327284@163.com (Y.C.); geruidong@126.com (R.G.); 3Wuhan Second Ship Design and Research Institute, Wuhan 430205, China; zhangfeiran1991@foxmail.com

**Keywords:** soft robotic suit, ankle assistance, powered planter flexion, iterative learning control

## Abstract

Concerning the biomechanics and energy consumption of the lower limbs, a soft exoskeleton for the powered plantar flexion of the ankle, named A-Suit, was developed to improve walking endurance in the lower limbs and reduce metabolic consumption. The method of ergonomics design was used based on the biological structures of the lower limbs. A profile of auxiliary forces was constructed according to the biological force of the Achilles tendon, and an iterative learning control was applied to shadow this auxiliary profile by iteratively modifying the traction displacements of drive units. During the evaluation of the performance experiments, four subjects wore the A-Suit and walked on a treadmill at different speeds and over different inclines. Average heart rate was taken as the evaluation index of metabolic consumption. When subjects walked at a moderate speed of 1.25 m/s, the average heart rate *H*_av_ under the Power-ON condition was 7.25 ± 1.32% (mean ± SEM) and 14.40 ± 2.63% less than the condition of No-suit and Power-OFF. Meanwhile, the additional mass of A-Suit led to a maximum *H*_av_ increase of 7.83 ± 1.44%. The overall reduction in *H*_av_ with Power-ON over the different inclines was 6.93 ± 1.84% and 13.4 ± 1.93% compared with that of the No-Suit and Power-OFF condition. This analysis offers interesting insights into the viability of using this technology for human augmentation and assistance for medical and other purposes.

## 1. Introduction

Lower limb exoskeletons are developed to reduce the metabolic consumption of human walking or running and are widely used in rehabilitation and strength enhancement [1]. According to the stiffness of the robot’s main components, the lower limb exoskeleton can be divided into two types: rigid and soft exoskeletons. The traditional rigid exoskeleton robots imitate the biological structure of human lower extremities, and the rigid linkages and components are taken as the main mechanical body. They can provide gait correction for patients with dyskinesia and walking assistance for healthy persons (BLEEX [2], HAL System [3], EKSO [4], and ReWalk [5]). However, the additional mass of these lower extremities and the axis misalignment of the human–machine system after wearing can easily lead to the increment of additional torques, disorders of natural gait, and dislocations of the auxiliary assistance [6,7]. In order to overcome the shortcomings of the rigid exoskeleton mentioned above, many scholars have studied soft exoskeletons that are more comfortable to wear (SEU-EXO [7], Exosuit [8], Myosuit [9], etc.), namely the soft power assistance system, which can assist the lower limbs in the state of walking or running [10]. The flexible driving units, such as the Bowden line, pneumatic muscle, and airbag, are applied to transmit the auxiliary forces and moments to the hip, knee, or ankle joints [11]. It is easy for the soft exoskeleton to adapt to the anatomical differences of different populations and the physiological changes of different movement modes. The tension parallel to human muscles or tendons can be compensated, and the wearer of the soft exoskeleton can walk more naturally.

The soft power assistance system supported by skeletons around the lower limb transfers the auxiliary forces through the flexible elements distributed along the skin’s surface to the single or multiple joints and reduces the loads and metabolic consumption of the corresponding flexor/extensor muscles. The Biodesign Lab at Harvard developed a soft exosuit using Bowden cable for the assistance of hip and/or ankle movement, which made great improvements [8,12,13,14]. The Stanford Research Institute Robotics group developed a soft robotic suit (Super-Flex) for assisting ankle plantar flexion through a twisted-string actuator anchored on the back of the lower limb [15]. A bioinspired design approach was used in a rehabilitation device, where four Mckibben PAMs were distributed to mimic the muscle-tendon ligament structure for assisting the movements of the ankle joint [16]. A soft inflatable robotic boot called ExoBoot was developed to assist the plantar flexion of the ankle using a pneumatic interference actuator [17]. A quasi-passive soft robotic suit (XoSoft) used two passive elastic elements connected in series with electromagnetic clutches for improving the ground clearance and gait symmetry of stroke patients [18]. The passive robotic suit ExoBand, with a total weight of 645 g, could reduce the metabolic cost of walking at 1.10 m/s by an average 3.30% without external power [19]. 

The relationship between the assistance magnitude and the metabolic cost of walking was isolated and characterized while also examining changes to the wearer’s underlying gait mechanics [20]. The contour of the auxiliary force was designed to supply assistance to the flexion movements of the hip joint based on the maximum/minimum hip angle and the corresponding time when the heel hits the ground [13]. Furthermore, according to the individual differences in walking, an individualized control strategy was proposed to adjust the control parameters of an assistive device and minimize the metabolic cost of walking using Bayesian optimization [14]. Two auxiliary profiles for walking and running were improved according to the biological hip moment and muscle-driven simulations, respectively [8]. Based on the kinematics and the ground-supporting force data collected by the VICON system and a force plate, the biological moment curves of the hip, knee, and ankle joints in the sagittal plane were obtained by the standard inverse dynamics method [21]. Based on the measurement experiments, a human musculoskeletal model of equal proportions was established to study the influence of external equipment on the lower limb muscle and metabolic consumption under loading and slope walking conditions [22]. The gait analysis affected by the external force and the activation degree of the hip flexors was discussed with the help of the soft exoskeleton H-Suit [23,24].

Bowden cable, consisting of steel cable and sheath, is widely used in soft robotic suits due to its advantages of flexibility, safety, and lightness. The distribution path of the Bowden cable can be planned according to the actual available space, which benefits from its own flexibility. Besides, the power can be transmitted over long distances, and the location of the drive unit is not limited. Benefitting from the above advantages, the ergonomics design of robotic suits focuses more on how to match the physiological structure of the limbs and improve human–machine compatibility. In order to achieve accurate controls of position, velocity, and auxiliary forces, a hierarchical controller was used to compensate for friction and backlash in the transmission by an adaptive paradigm [25]. A more pragmatic approach adopted in the robotic suits mentioned above is to use a high-gain position or velocity loop cascaded with the force controller. A force-based position feedback controller was applied to track the desired assistance profile [12,26]. The force feedback measurement is used to update the cable displacement, only once per step, to maintain consistent force application. An alternative approach for indirect force control comprises inner-velocity and outer-force control loops [27,28,29]. The admittance control strategy was used in the two approaches above, where an experimentally identified human-suit stiffness and a desired virtual admittance are needed. Additionally, an iterative learning control strategy is adopted to approach the preset contour of auxiliary forces for reducing the errors generated due to the difference between the wearing position and the biological features of the different users [30].

Understanding how those devices affect the metabolic consumption of human bodies is fundamental for quantifying their benefits and drawbacks, assessing their suitability for different walking terrains, and guiding continuous data-driven design refinement. We present a soft exoskeleton named A-Suit for the plantar flexion of the ankle, using the method of ergonomics design based on the biological structures of the lower limbs. A profile of auxiliary forces was determined and the iterative learning control was applied to shadow this auxiliary profile through iteratively modifying the traction displacements of the drive units. The average heart rate was taken as the evaluation index of metabolic consumption. The changes of heart rates under the different walking speeds and inclines were analyzed and are herein discussed comprehensively. The ergonomics design and the evaluation method of assistance performance are the innovative technologies in this research, in which the wearable comfort and assistance strategy may find continuous improvement over future daily use. The analysis in this paper offers interesting insights on the viability of using this technology for human augmentation and assistance for medical and other purposes.

## 2. Materials and Methods

### 2.1. Ergonomics Design of A-Suit

#### 2.1.1. Optimization of Anchor Locations

The ankle joint can be simplified as a spherical joint with three rotational degrees of freedom, which perform movements of plantar flexion/dorsiflexion (PL/DO), adduction/abduction (AD/AB), and introversion/eversion (IN/EV), respectively. Research shows that the movements of AD/AB and IN/EV play a significant role in keeping balance and adjusting the direction of movements, and the corresponding ranges of motion are relatively small (less than 5°) [31,32]. PL/DO, with the peak torque of the ankle, has a larger range of motion and longer duration. The energy consumption of plantar flexion is also significantly greater than that of dorsiflexion. Thus, plantar flexion is taken as the assistance movement.

The assistance of the soft exosuit (A-Suit) for the plantar flexion consists of an exosuit body, a drive unit, and Bowden cables; a sketch of A-Suit is shown in Figure 1a. The anchors used to fix the Bowden cables are sutured on the calf garments and the straps of the boot covers. The auxiliary forces are transmitted to the ankle joint through the Bowden cable, where the lower ends of the sheath and the steel cable are fixed at the upper and lower anchors, respectively. The lower anchor, denoted by A, is located on the back of the heel, and upper anchor B on the gastrocnemius surface, as shown in Figure 1b. When the steel cable is pulled by the drive unit, the tension forces *F*_t_ from lower anchor A to upper B are produced to generate the assistance for the plantar flexion. In order to ensure the efficiency and stability of the assistance process, the rated output of the drive unit and the smoothness of auxiliary forces should be taken as the objective function to solve the optimal anchor locations.

The biological structures of the gastrocnemius and heel are approximately arc-shaped. The upper/lower anchors are located on the two arcs, respectively. The optimization of anchor locations can be transformed into a geometric solution, and the optimization model can be established for the standing posture. The mathematical model of the anchor locations establishes and solves the torque arm *d* of the auxiliary forces *F*_t_ and the initial distance *l*_0_ corresponding to different locations of the upper/lower anchors, as shown in Figure 1c. The mathematical model and results are presented in Appendix A. When the torque arm *d* reaches a maximum *d*_max_ (*d*_max_ = 143.2 mm), the optimal anchor locations in the coordinate *x*_a_*o*_a_*y*_a_ are obtained. Furthermore, the initial length *l*_0_ between the optimal locations (*l*_0_ = 180.9 mm) can meet the requirements of the space of load cells and quick-release connectors.

#### 2.1.2. Ergonomics Design of the A-Suit System

The A-Suit system constitutes the drive unit, Bowden cables, anchor parts, the flexible exosuits and the gait detection unit, as shown in Figure 2. The two ends of the sheath are respectively fixed on the drive unit and the upper anchor parts. Similarly, the two ends of the steel cable are fixed on the capstan of the drive unit and the lower anchor. The traction ability of the drive unit is achieved by the axial movements of the steel cable along the sheath. The external mechanical energy is transmitted to the upper/lower anchors to replace part of the required metabolic energy. A-Suit is worn on the surface of the lower limbs, and the additional mass inevitably leads to increased increments of metabolic consumption. Furthermore, the metabolic consumption raises sharply with increments at the distal limb. Therefore, the drive unit and the battery are fixed on the back and front of the waist, respectively, to reduce the additional mass on the calf and foot as much as possible. The detailed structure of the flexible exosuit was optimized to improve its strength and reduce the total mass. The mass distribution along the lower limbs is shown in Figure 2e.

Based on the selected anchor positions, the connecting components with a butterfly shape for fixing the sheath and the steel cable are designed and manufactured with nylon materials and 3D printing technology. The metal bases are sewed at the anchors of the exosuit, and the components are connected to the metal base by screws. The exosuit, with a good coating property and high wearable comfort, consists of the bottom garment, calf garment, boot cover and Velcro. The bottom garment is skin-friendly and can be worn directly on the lower limbs. The calf garment is made of breathable mesh fabric and tightly bound to the calf surface by Velcro. Concerning the structure of the military boot, the boot cover consists of nylon fabric belts which provide the installation location for the connecting components and IMUs.

The gait detection unit is used to measure the postures of the lower limbs and soles to provide the starting/halt time for the auxiliary forces. IMUs are fixed on the surfaces of the shanks and insteps, respectively. The attitude angles of the IMUs at the sagittal plane are collected and the angles of the ankle joints and the soles are obtained at the same time. A reference IMU is added on the anterior abdomen of the trunk. When humans walk uphill, downhill, on stairs, or other road conditions, the trunk generally maintains a stable vertical or forward posture. The walking environment can be predicted by the reference IMU and the characteristics of the ankle angles to adjust the traction displacement and assistance profile in time.

The drive unit, which consists of two identical traction devices and a control unit, is used to pull the steel cables of the Bowden cables and assist the plantar flexions of the left and right ankles, respectively. The traction device is composed of a DC motor (RE40, Maxon motor Inc., Sachseln, Switzerland), a reducer (transmission ratio 13:1), a driver (Epos4, Maxon motor Inc., Switzerland), and a capstan. The capstan base is installed on the flange of the reducer by means of the screwed connection. The capstan is directly mounted on the output shaft of the reducer. The protective shell and the capstan base form a closed cavity for the steel cable. The steel cable is wound in the outer grooves of the capstan. The upper end of the sheath is fixed on the capstan base. When the capstan rotates anticlockwise, the steel cable is twined in the outer groove and moves along the axis of the sheath to draw the lower anchor and generate plantar flexion assistance, as shown in Figure 3. The kinematics of the drive unit are determined by the control strategy. The range of motion of the capstan is 300°, and the corresponding traction displacement is 210 mm. Additionally, the maximum traction force is 120 N, and the maximum speed of the cable can reach 1.5 m/s.

A prototype of the drive unit was developed, and the total weight of A-Suit is 5.2 kg. The load cells are connected in series between the Bowden cable and lower anchors to measure the auxiliary forces in real-time. Based on the gait information, an expected profile of auxiliary forces can be applied to assist with ankle movement through controlling the traction displacements. If the drive unit produces a dangerous tension, A-Suit can indeed damage the user’s lower limbs and body balance [33]. Hence, safety was taken as the primary condition that must be satisfied. The maximum tension and maximum speed are set in the control software. When one of them exceeds the maximum value, the motor will be automatically powered off, and the steel cable can be pulled out under the action of a small external force to reduce the constraint and injury to the ankle joint. Additionally, a limit block is added in each traction unit to limit the rotation angle of the capstan and the traction distance of the steel cable.

### 2.2. Assistance Strategy of A-Suit

Based on the biomechanical characteristics of the ankle joint, the ideal assistance profile was established. Furthermore, the natural and elastic displacements of the Bowden cable were calculated for human walking. The ideal assistance profile was achieved by adopting iterative learning control (ILC).

#### 2.2.1. Profile of the Auxiliary Force

The single gait cycle *T* starts when the heel hits the ground and ends when the heel hits the ground for a successive time [34]. The cycle *T* can be divided into the stance phase and the swing phase. When the foot is completely off the ground, this is regarded as the swing phase of the gait cycle. The stance phase begins with the heel hitting the ground and ends with the toe leaving the ground. The angle *φ*_f_ of the foot at the sagittal plane is measured by the IMU stuck to the instep, and the ankle angle *θ* is the difference between the angles of the foot and instep. The two angles in a single cycle *T* are measured by the gait detection unit at a walking speed of 1.25 m/s, as shown in Figure 4a. The incline slope of the walking environment can be detected by the angle *φ*_f_. Meanwhile, the traction displacements of the drive unit are directly influenced by the ankle angle *θ*.

The plantar flexion mainly occurs in the stance stage, where the biological force of the ankle, especially the Achilles tendon, should be analyzed. The biological force *F*_bio_ has a triangle shape and reaches its maximum during the max. Dorsiflexion propels the body upward and forward. The profile of the auxiliary force *F*_a_ starts at 45%*T*, and ends at 67%*T* (*T*_a_ ∈ [45%*T*, 67%*T*]). The force *F*_a_ reaches a maximum from 47%*T* to 54%*T,* where *F*_a,max_ = 80 N, as shown in Figure 4b.

#### 2.2.2. Initial Displacement of the Bowden Cable

The natural displacement of the Bowden cable is used to track the changes in distance between the upper and lower anchors during normal walking. The reference coordinate *x*_d_*o*_a_*y*_d_ is established at the rotation center *o*_a_, and moves with the ankle. Additionally, the transverse axis *o*_a_*x*_d_ is always horizontal. The coordinate *x*_1_*o*_1_*y*_1_ is connected to the shank IMU, and the vertical axis *o*_1_*y*_1_ is parallel to the axis *o*_a_*y*_a_. Similarly, the coordinate *x*_2_*o*_2_*y*_2_ is fixed onto the instep IMU, and the horizontal axis *o*_2_*x*_2_ is parallel to the sole, as shown in Figure 5. Since the above coordinate systems are mainly used to calculate the postures of the ankle and foot, the corresponding origins can be superimposed, making it easier to express the included angles among the coordinates. The shank angle *α* relative to the vertical axis *o*_a_*y*_d_ is measured by the shank IMU, and the instep angle *β* is relative to the vertical axis *o*_a_*x*_d_ via the instep IMU. *α* and *β* are positive when the coordinates *x*_1_*o*_1_*y*_1_ and *x*_2_*o*_2_*y*_2_ rotate counterclockwise. The included angle between the axes *o*_a_*y*_1_ and *o*_a_*x*_2_ is defined as the ankle angle *θ*. Based on the superimposed coordinate systems, the ankle angle *θ* can be obtained as follows:(1)$$\theta =\theta_{0}+\alpha -\beta$$
where *θ*_0_ = π/2.

The lower/upper anchors A and B are fixed to the back of the heel and the gastrocnemius surface, respectively. The vector ***o*_a_*A*** moves with the coordinate *x*_2_*o*_2_*y*_2_ and vector ***o*_a_*B*** with the coordinate *x*_1_*o*_1_*y*_1_. The included angle *φ*_a_ between the two vectors can be acquired as the following:(2)$$\varphi_{a}=\varphi_{a0}-\alpha +\beta$$

Substituting Equation (1) into Equation (2), the function of *φ*_a_ with respect to *θ* can be solved:(3)$$\varphi_{a}=\varphi_{a0}+\frac{\pi }{2}-\theta$$

The natural displacement *x*_n_ of the drive unit is the change in the length *l*_AB_ of AB relative to its initial length *l*_0_, which can be calculated as follows:(4)$$x_{n}=l_{0}-l_{AB}$$

According to the cosine theorem of trigonometric functions, *l*_AB_ can thus be achieved:(5)$$l_{\mathrm{AB}}=\sqrt{{\left|o_{a}A\right|}^{2}+{\left|o_{a}B\right|}^{2}-2\left|o_{a}A\right|\cdot \left|o_{a}B\right|\cdot \cos\varphi_{a}}$$

A volunteer wears A-Suit and walks on a horizontal treadmill at a moderate speed of 1.25 m/s. The angles *α* and *β* are collected by the shank and instep IMUs. Then, angles *θ* and *φ*_a_ are obtained. The natural displacement *x*_n_ within the gait cycle *T* can be achieved, as shown in Figure 6.

The A-Suit system will generate obvious elastic deformation under the traction force *F*_t_. The transmission efficiency of the auxiliary force is restricted by the total stiffness *k*_total_ of the system. *k*_total_ is mainly composed of the stiffness of the A-Suit body, local soft tissues, and the steel cable. The method of experimental measurement is adopted to calculate the stiffness *k*_total_ [26]. The traction force *F*_t_ is continuously applied to the ankle joint from 0 N to 120 N through the drive unit at the standing posture. Meanwhile, the ankle produces the corresponding resistance and keeps relatively stationary. The elastic displacement *x*_e_ of the Bowden cable was measured and recorded by a displacement sensor. Based on the collected data of *x*_e_, four real curves were formed and plotted, shown in Figure 6a. The fitting curve of the forces and displacements is the average value of the four measurement curves. The stiffness *k*_total_ was calculated with OriginPro software. Hence, the expression of *k*_total_ can be fitted:(6)$$k_{\mathrm{total}}=0.8+0.019x_{e}$$

The desired displacement *x*_d_ of the Bowden cable under the auxiliary profile of *F*_a_ consists of the displacements *x*_n_ and *x*_e_:(7)$$x_{d}=x_{n}+x_{e}$$

Based on the auxiliary profile of *F*_a_ and the stiffness *k*_total_, the desired displacement *x*_d_ during a walking speed of 1.25 m/s is calculated as shown in Figure 6b.

#### 2.2.3. Adaptive Detection Algorithm

The A-Suit system is a typical human–machine coupled system. The corresponding performance is directly restricted by the accuracy of the gait information. An adaptive gait detection algorithm was proposed for changing the assistance ranges and determining the trigger and halt times of the drive units according to the gait cycles at different walking speeds. The gait information, especially the foot angle *φ*_f_, is detected by IMUs. The trigger time *T*_tr_ and the halt time *T*_ha_ can be obtained dynamically by analyzing the changes of the angles *φ*_fe_ and *φ*_min_.

The collected data *φ_i_* of the angle *φ*_f_ are temporarily stored in the array called *φ*-queue. When *φ_i_* = 0°, the current datum *φ_i_* will be compared with the previous three data. If the previous angles are all greater than *φ_i_*, the current datum *φ_i_* will be defined as *φ*_fe_, when the angle is equal to *φ*_fe_, it is about 45%*T* at this time, and the drive units will be triggered to drag the steel cable immediately. During a gait cycle *T*, the current datum *φ_i_* which is less than 0° will be compared with the previous *i*-1 angles. When the datum *φ_i_* is less than all *i*-1 angles, the minimum *φ*_min_ can be acquired, when the angle is equal to *φ*_min_, it is about 67%*T* at this time and the drive units will be stopped at this time. When the datum *φ_i_* on the rising edge of the angle *φ*_f_ (Figure 4a) equals 0°, *φ_i_* is defined as *φ*_re_ and the *φ*-queue is refreshed. The criterion for *φ*_re_ is similar to that of *φ*_fe_. The criteria for the above three angles are as follows:(8)$$\varphi_{i}=\{\begin{array}{c} \begin{array}{l} \varphi_{\mathrm{fe}}\\ \varphi_{\min}\\ \varphi_{\mathrm{re}} \end{array} \end{array}\begin{array}{c} \left(\varphi_{i-3}>\varphi_{i-2}>\varphi_{i-1}>\varphi_{i}\right)\\ \left(\varphi_{1},\ldots ,\varphi_{i-1}>\varphi_{i}\right)\\ \left(\varphi_{i-3}<\varphi_{i-2}<\varphi_{i-1}<\varphi_{i}\right) \end{array}$$

When the foot angle *φ*_f_ reaches *φ*_fe_, the drive unit begins to move, and the steel cable is pulled to produce the desired profile of the auxiliary force *F*_a_. The steel cable will be released quickly when the angle *φ*_f_ reaches *φ*_min_. 

### 2.3. Control Strategy of the A-Suit

Due to the hysteresis effect, the dynamic asymmetry of the one-way transmission and nonlinear coupled problems of Bowden cables [35], delays of dynamic response and unknown jitters may occur during the assistance process. It is difficult to realize the high-accuracy force-position control of the drive unit. For solving the above problems, an ILC strategy was proposed according to the periodicity of human gait [36,37]. An iterative learning law was constructed to correct the control variable *x*_d,*k*_(*t*) at current moments by continuously learning the historical errors of the output forces [38]. The traction displacement *x*_d,*k*_ of the drive unit at the *k*_th_ gait cycle is taken as the input and the auxiliary force *F*_a,*k*_ as the output. The displacement *x*_d,*k*_ is adjusted constantly based on the historical errors. The auxiliary profile of the traction force *F*_a,*k*_ can be converted quickly to the ideal profile of *F*_a_.

The key of ILC is to construct a learning law to obtain the desired control variable *x*_d_(*t*) [39,40]. Under the effects of *x*_d_(*t*), the system output *F*_a,*k*_(*t*) can approach the desired profile of *F*_a_(*t*) as accurately as possible within the auxiliary range *T*_a_. The state-space representation of the auxiliary system can be obtained through the system identification toolbox of MATLAB.
(9)$$\left[\begin{array}{c} y_{1,k}\left(t+1\right)\\ y_{2,k}\left(t+1\right) \end{array}\right]=\left[\begin{array}{cc} 0.757 & 0.371\\ -0.925 & 0.61 \end{array}\right]\left[\begin{array}{c} y_{1,k}(t)\\ y_{2,k}(t) \end{array}\right]+\left[\begin{array}{c} 0.014\\ 0.036 \end{array}\right]x_{d,k}(t)$$
(10)$$F_{a,k}(t)=\left[\begin{array}{cc} 33.46 & 2.84 \end{array}\right]\left[\begin{array}{c} y_{1,k}(t)\\ y_{2,k}(t) \end{array}\right]$$
where *y_i_*_,*k*_(*t*) (*i* = 1,2), *x*_d,*k*_(*t*), and *F*_a,*k*_(*t*) are the state variables, controlled input variable, and output variable of the auxiliary system at the *k*_th_ iteration, respectively. 

Based on the desired profile of *F*_a_(*t*) (Figure 4b), the corresponding approximate expression can be fitted by the software of OriginPro:(11)$$F_{a}(t)=\sum_{i=0}^{5}A_{i}{\left(t\right)}^{i}\begin{array}{c} \left(t\in T_{a}\right) \end{array}$$
where *A_i_* is the polynomial coefficient and shown in Table 1.

The second-order P-type ILC was proposed to improve the convergence rate of the system, as shown in Figure 7. The iterative error *e_k_*(*t*) and the learning law are as follows:(12)$$e_{k}(t)=F_{a}(t)-F_{a,k}(t)$$
(13)$$x_{d,k+1}(t)=x_{d,k}(t)+Le_{k}(t)$$
where *L* is the learning gain.

If *x*_d,*k*_(0) = *x*_d_(0) (*k* = 0,1,2…), *F*_a,*k*_(*t*) generated under the effects of ILC will eventually converge to *F*_a_(*t*).
(14)$$\underset{\infty }{\lim}F_{a,k}(t)=F_{a}(t)\;(t\in T_{a})$$

## 3. Results

Since efficient and accurate assistance for the plantar flexion is the primary goal of the A-Suit, performance experiments were designed. When walking speed exceeds 1.5 m/s, the optimum form of human motion will switch from walking to running. The maximum walking speed during the experiments was 1.5 m/s. Four subjects (male, average age: 25, average height: 175 cm, average weight: 77 kg) wore the A-Suit and walked on the treadmill at the speed of *V_m_* (*m* = 1, 2, 3). The actual performance of the assistance planning and the control strategy of ILC was verified by comparing the assistance profile of *F*_a_ with the actual profile of *F*_a,*k*_ collected by load cells. The learning gain *L* of ILC was optimized and obtained through the numerical simulation in Simulink (of MATLAB) to improve the convergence rate. The maximum allowable output error *e*_max_ of *F*_a,*k*_(*t*) is 5 N. The main parameters are listed in Table 2.

There were three experiments for the performance evaluations: (I) wear A-Suit and turn on the power switch (Power-ON); (II) wear A-Suit and turn off the power switch (Power-OFF); and (III) do not wear A-Suit (No-Suit). A subject performs each experiment by walking on the treadmill for 10 min, as shown in Figure 8a. The four subjects were informed that they could not eat and not exercise 3 h before the tests to prevent further variability in energy consumption that could affect the experimental results [41,42]. A group of experiments were carried out each day and all evaluations were completed within 6 days. The subjects rested for 20 min before the test to keep their resting metabolic level and walked in the state of Power-ON for 10 min. In addition, the walking speed changed every two days from low to high. 

The average heart rate *H*_av_ and oxygen consumption had a positive correlation during the load exercises [43]. A heart rate meter (YX306, Yuwell Inc., Danyang, China) was used to collect the heartbeat rate of the subjects during the experiments. Then the actual assistance performance of A-Suit was verified by comparing the changes in *H*_av_ [23,44]. In the process of the assistance experiments, the auxiliary forces *F*_a,*k*_ were collected by the load cells, and the traction displacement *x*_d,*k*_ of the Bowden cable was calculated by the encoders of the drive units. Each type of experiment was conducted twice at the speed of *V*_m_. All experimental results are the average of the collected data used to evaluate the performance of A-Suit.

The preset displacement *x*_d,0_ was taken as the initial input of the control unit. The control variable *x*_d,*k*_(t) of ILC was constantly modified via iteratively learning the errors *e_k_*(*t*) of *F*_a,*k*_. At the *k*_th_ gait cycle, the input displacement *x*_d,*k*_ and the output force *F*_a,*k*_ were measured and recorded in the control unit. The results of the experiments for the moderate speed *V*_2_ = 1.25 m/s (experimental data of subject No.1) are shown in Figure 9a. The iterative force *F*_a,*k*_ at the *k*th gait cycle approaches the desired profile of *F*_a_ gradually between the 11th~15th iterations, and the relative errors between *F*_a,*k*_ and *F*_a_ are limited to the maximum allowable output error *e*_max_. The actual input displacement *x*_d,*k*_ was continuously changed under the effect of ILC. In the assistance stage, the real displacement *x*_d,*k*_ of the drive unit can allow smooth changes during walking assistance, and the errors of *F*_a,*k*_ occur mainly due to the stiffness discrepancies and the elastic hysteresis of the exosuit body. The root mean square error (RSME) of *k*th iteration decreased with the number of iterations, as shown in Figure 9b. The force *F*_a,*k*_ was able to track the profile of the desired auxiliary force *F*_a_ quickly. From the perspective of usability, the equipment can understand the movement characteristics of the lower limbs, and the movement fluency between the human and the machine is continuously improved. The subjects obviously felt the process to be labor-saving.

The heart rates of the subjects rose with the increment of walking time. When subjects walked at the three different speeds, their average heart rates *H*_av_ were calculated based on the experimental data collected during the tests, as shown in Figure 10a. Solid curves indicate actual changes in *H*_av_ during the three walking conditions of moderate speed 1.25 m/s, while error bars indicate the standard error of mean (SEM). Reported values and measurements from here onwards, in both graphs and text, are presented as mean ± SEM. When the A-Suit was powered on, the heart rates *H*_av_ declined massively, and the metabolic consumption also decreased greatly. However, the portable A-Suit generates additional mass and leads to fair increments in heart rate *H*_av_, such as the 7.15 ± 1.44% increment when under the conditions of Power-OFF at *V*_2_ = 1.25 m/s, as shown in Figure 10b. *H*_av_, in the state of Power-ON, decreased by 6.84 ± 1.41%, 7.25 ± 1.32%, and 6.25 ± 1.73%, compared with that of No-Suit at speed *V*_m_, respectively. Heart rate *H*_av_ under Power-ON at *V*_1_ = 1.0 m/s can be reduced by 14.67 ± 1.21% compared with Power-OFF. The performance of A-Suit can improve with an increase in walking speed. However, the mass of A-Suit also consumes more metabolic energy. The subjects could clearly feel the force exerted by the external device on the ankle. The control strategy of ILC can quickly track the gait of subjects with the increase in walking time. Therefore, A-Suit can assist the ankle in the expected range according to the current angle *φ_k_*. The actual profile of *F*_a,*k*_ could rapidly and accurately approach the desired profile of *F*_a_ via the method of ILC.

The experiments on the flat treadmill do not fully reveal the real assistance performance of A-Suit. It is also necessary to evaluate the performance of A-Suit when on two typical slopes, such as uphill (+10°) and downhill (−10°). The heart rate *H*_av_ of each individual for the slope conditions are exhibited in Figure 11. The means of the heart rates *H*_av_ under No-Suit and Power-OFF conditions decrease from uphill to downhill conditions. The heart rates *H*_av_ declined markedly when A-Suit was turned on. The overall reduction of *H*_av_ for Power-ON was 6.93 ± 1.84% and was 13.4 ± 1.93% when compared with No-Suit and Power-OFF conditions. The performances under different inclines were roughly equal with no specific regularity. Therefore, A-Suit can be applied to walking uphill and downhill.

## 4. Discussion

The methods of ergonomic design were proposed in my previous research [11,23,24], including the optimization of anchor points, comfort evaluation, analysis of biomechanical characteristics of the lower limb, etc. Additionally, our team has obtained two national patents for soft robotics [45,46] and developed three types of exosuits for the hip and/or ankle joints. Compared with the development of the existing soft assistance systems [8,16,19], the design process is here presented in more detail. The A-Suit system can guarantee the improvement of assistance performance in the future.

According to the periodicity of the human gait, we propose the strategy of ILC to overcome the hysteresis effect and the dynamic asymmetry and nonlinear coupled problems of Bowden cables. The subjects really feel that the external forces can quickly and softly follow those of the gait and the lower limbs. Although Quinlivan [12], Schiele [20], and Xiloyannis [27] applied admittance control strategies, ILC can also achieve considerable performance and provide a backup method for the soft assistance suits. Meanwhile, this method can deal with unknown repetitive interference and avoid the hysteresis effect caused by traditional closed-loop controls.

The metabolic consumption of the human body is calculated by measuring the proportion and quantity of carbon dioxide and oxygen in the exhaled gas, which is used to evaluate the performance of the power assistance system. However, the gas detector is easily affected by the ambient air and basic metabolism, and only measures the overall metabolic consumption over a long duration, making it difficult to measure the real-time values. The gas detection equipment, especially the portable ones, are generally expensive. Heart rate, which directly reflects the changes of metabolic level [41,43,44], can be measured and displayed in real-time. Heart rate detection is a mature technology, and the specialized equipment to measure it is cheap. In addition, intelligent bracelets and watches can generally detect heart rate. Hence, the average heart rate is taken as the evaluation index of assistance performance.

With incremental changes in walking steps, A-Suit can quickly follow the gait characteristics of different subjects. Similarly, with the increase in test duration and familiarity with the assistance system, subjects adjusted their gaits to adapt to the movements of the traction units. The cooperation between the traction movements and the intended movements of the lower limbs can reduce the biological force of ankle plantar flexion and save on the metabolic consumption of the human body. The subjects’ behaviors during the performance experiments are consistent with the research results in the literature [47]. The human body can reduce the total energy demand for different road conditions by adjusting pre-controlled momentum and extracting the positive energy from the external assistance equipment.

## 5. Conclusions

The exosuit is mainly composed of flexible fabric, and it does not affect the natural activities of the lower limbs. A-Suit can be used in daily life due to its portability and availability. During the experiments of performance evaluation, four subjects wore A-Suit and walked on a treadmill at different speeds and inclines. When subjects walked at a moderate speed of 1.25 m/s, the average heart rate *H*_av_ for Power-ON was 7.25 ± 1.32% (mean ± SEM) and was 14.40 ± 2.63% less than that for the No-suit and Power-OFF condition. Additionally, A-Suit generates an additional mass for the user and leads to the maximum increase in *H*_av_ of 7.83 ± 1.44%. The overall reduction in *H*_av_ for Power-ON over the different inclines was 6.93 ± 1.84% and 13.4 ± 1.93% compared with the No-Suit and Power-OFF condition. The experimental results show that although the driving unit increases the load on the human body, A-Suit can still produce a significant assistance effect via reasonable weight reduction and the control strategy.

Future studies will include further weight reductions by optimizing the mechanical structure of the suit, with the aim of achieving greater energy expenditure reduction. In addition, exploration of the best assistive force profiles for different walking patterns is necessary. Based on this, our final goal is to develop an effective, lightweight, completely wearable robotic suit that improves walking endurance for healthy people.

## Figures and Tables

**Figure 1 micromachines-13-01114-f001:**
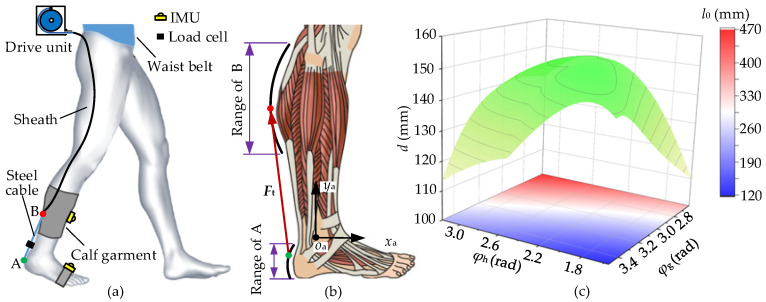
Optimization of anchor locations. (**a**) A-Suit, for plantar flexion, consists of an exosuit body, a drive unit, and Bowden cables. (**b**) Biological structures of the gastrocnemius and heel are measured and shaped, where the anchors A and B move along the arcs. (**c**) The torque arm *d* and initial length *l*_0_ are calculated, with the changes of the angles *φ*_h_ and *φ*_g_.

**Figure 2 micromachines-13-01114-f002:**
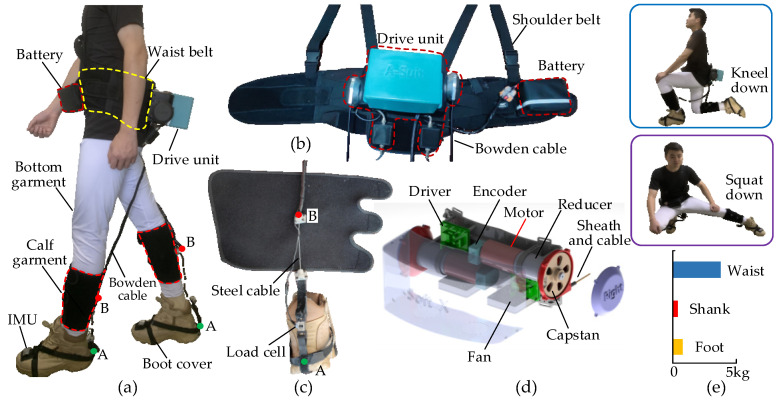
Design and actuation of A-Suit. (**a**) A-Suit system is composed of the drive unit, Bowden cable, anchor parts, flexible exosuit, and gait detection unit. (**b**) The drive unit and battery are fixed to the waist belt and placed on the back and front of the body. (**c**) Anchors A and B are sutured on the calf garment and boot cover, and the load cells are connected in series between the two anchors. (**d**) The drive unit consists of two traction devices and the steel cables are pulled and released by the capstan. (**e**) The A-Suit doesn’t affect the natural movements of the human body, and most of the weight is distributed around the waist.

**Figure 3 micromachines-13-01114-f003:**
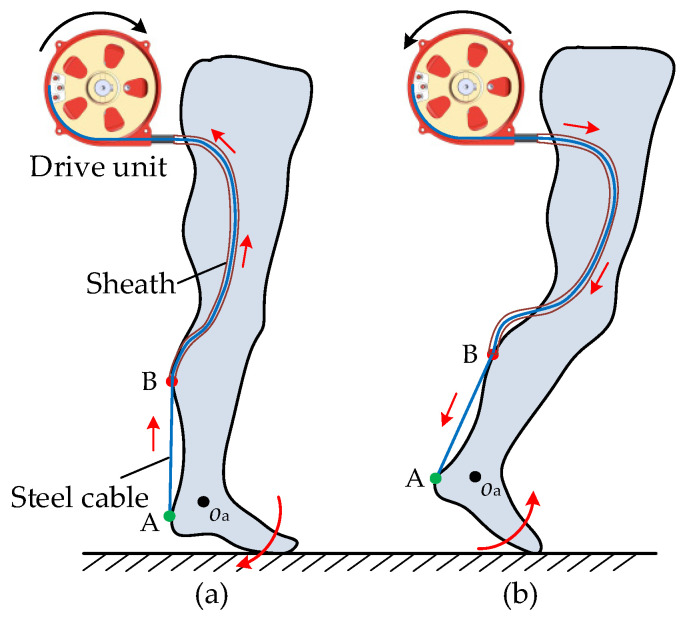
Kinematics of the drive unit. (**a**) Contraction of the steel cable at the plantar flexion of the ankle. (**b**) Release of the steel cable at the prophase of the swing phase.

**Figure 4 micromachines-13-01114-f004:**
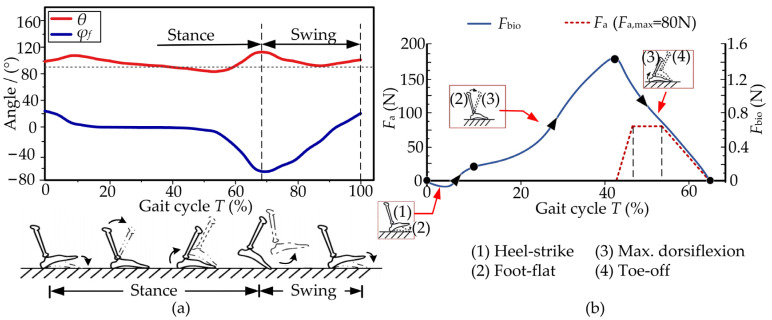
Biomechanical characteristics of the ankle joint. (**a**) The angles *φ*_f_ and *θ* are measured by the gait detection unit. The external assistance mainly happens in the stance phase. (**b**) The desired auxiliary force *F*_a_ is designed from the biological force *F*_bio_ of the ankle joint during the four actions (male; weight: 78 kg; height: 175 cm; walking speed: 1.25 m/s).

**Figure 5 micromachines-13-01114-f005:**
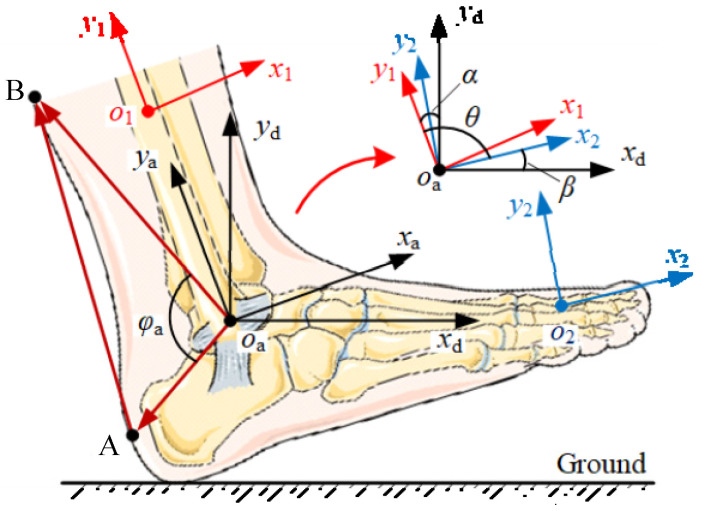
Coordinate systems established on the ankle.

**Figure 6 micromachines-13-01114-f006:**
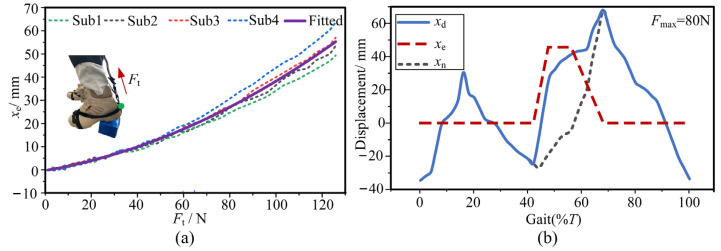
Desired displacement of the Bowden cable. (**a**) The total stiffness *k*_total_ is measured and fitted by the drive unit at the standing posture. (**b**) The desired displacement *x*_d_ is the sum of the displacements *x*_n_ and *x*_e_.

**Figure 7 micromachines-13-01114-f007:**
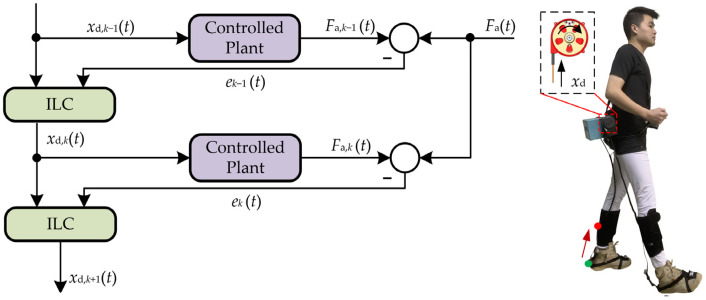
Algorithm flow of second-order ILC. The key of ILC is to construct a learning law to obtain the desired control variable *x*_d_(*t*), and approach the desired profile of *F*_a_(*t*) as accurately as possible.

**Figure 8 micromachines-13-01114-f008:**
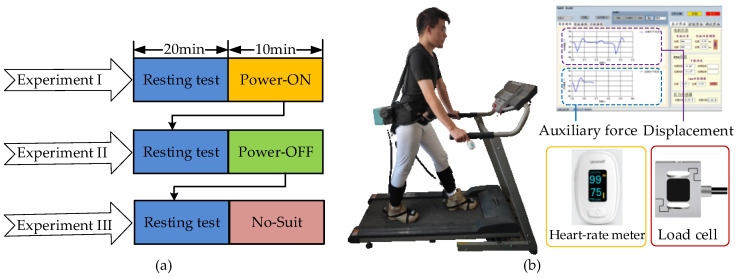
Scheme and platform of assistance experiments. (**a**) The volunteers rested for 20 min before the test and walked in the state of Power-ON, Power-OFF, and No-Suit for 10 min. (**b**) The volunteers walked on the treadmill at three different speeds and heart rate, auxiliary force, and displacement data were measured and depicted in the visual interface.

**Figure 9 micromachines-13-01114-f009:**
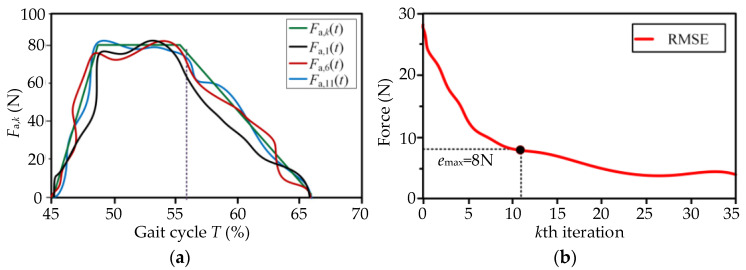
Output force *F*_a,*k*_ and input displacement *x*_d,*k*_ (*V*_2_ = 1.25 m/s on the treadmill). (**a**) The iterative force *F*_a,*k*_ at the *k*th gait cycle approached the desired profile of *F*_a_ gradually after 11th~15th iterations. (**b**) The force *F*_a,*k*_ can quickly track the profile of the desired auxiliary force *F*_a_.

**Figure 10 micromachines-13-01114-f010:**
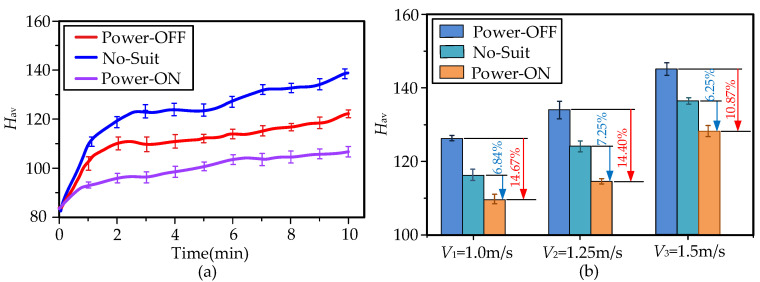
Heart rates *H*_av_ under different conditions. (**a**) The heart rates *H*_av_ were depicted in real time under three walking conditions of moderate speed 1.25 m/s (*p* = 0.015), while error bars indicate the standard error of mean (SEM). (**b**) The changes in *H*_av_ for the three walking speeds are presented clearly. The assistance of A-Suit significantly reduced the heart rates of the volunteers. Hence, it is verified that the A-Suit system can reduce human metabolic consumption.

**Figure 11 micromachines-13-01114-f011:**
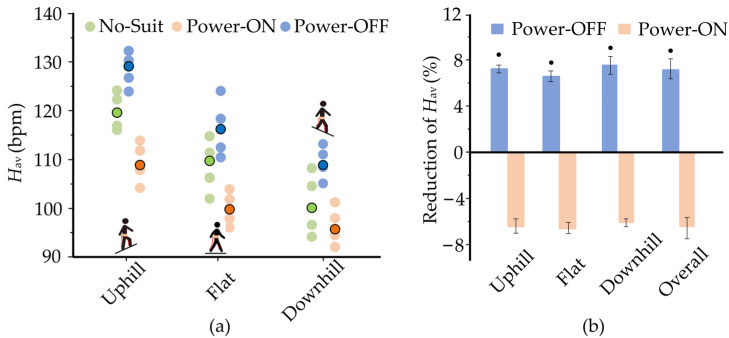
Changes in heart rates *H*_av_ over different inclines (*V*_2_ = 1.25 m/s). (**a**) The heart rates *H*_av_ of the four subjects over Uphill terrain (+10°), Flat, and Downhill terrain (−10°) are depicted clearly. Translucent circles are the values for each individual subject; opaque contoured circles indicate the mean over the subjects. There are no outliers identified by Thomson tau analysis. (**b**) A-Suit can be applied to walking uphill and downhill. The overall reduction of *H*_av_ for Power-ON is 13.4% ± 1.94% compared with that of Power-OFF. Black cycles indicate a significant difference from 0.

**Table 1 micromachines-13-01114-t001:** Parameters of the desired profile.

Params	Values	Params	Values
*A* _0_	−394.05	*A* _1_	−56.07
*A* _2_	3514.71	*A* _3_	−6837.22
*A* _4_	3797.91	*A* _5_	0.44

**Table 2 micromachines-13-01114-t002:** Parameters of performance experiments.

Params	Values	Params	Values
*V* _1_	1.0 m/s	*V* _2_	1.25 m/s
*V* _3_	1.50 m/s	*L*	2.26

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
