# Peer review of "Ergonomics Design and Assistance Strategy of A-Suit"

_micromachines, 2022, doi:10.3390/mi13071114_

Round 1

Reviewer 1 Report

Ergonomics Design and Assistance strategy of Ankle assiatance Exosuit

Dear authors

It is a complete work, which presents valuable results for the scientific community. It has satisfactory results that meet the objectives of the study. 

Here are some observations that could improve the content of the study. So that various technological areas understand the importance of your research.

·         Is it necessary to include the word “assistance” twice in the title? Also, the word “assistance” is misspelled in the title. ““Ergonomics Design and Assistance strategy of Ankle assiatance Exosuit”.

·         In the introduction you did not mention the innovative technologies in your design. And the benefits you get from using them.

·         You need to extend a little more the advantages of using soft exoskeletons. To support this work even more, it would be important to present more references, I leave a work that would be of great contribution to this article:

“Pérez Vidal, A. F., Rumbo Morales, J. Y., Ortiz Torres, G., Sorcia Vázquez, F. D. J., Cruz Rojas, A., Brizuela Mendoza, J. A., & Rodríguez Cerda, J. C. (2021, July). Soft exoskeletons: development, requirements, and challenges of the last decade. In Actuators (Vol. 10, No. 7, p. 166). MDPI.”

·         On lines 63-64, it mentions individualized control strategy for walk types. You can describe this idea in more detail, for example by mentioning the type of walk with the corresponding control used.

·         In lines 72 and 73 you mention the Bowden cable. You can complement the information on Bowden cables, mentioning the advantages of using them in robotics.

In the opinion of this reviewer, the work presented is valuable and worthy of being published as long as the observations mentioned above are addressed.

Author Response

Thank you very much for your valuable comments. Please refer to the document for all answers.

Reviewer 2 Report

This paper presents a soft exoskeleton for ankle assistance strategy.

The paper is interesting and presents a novel approach to ankle assisitng problem. However there are major improvement that has to be done according to my opinion to allow the publiction of this paper.

Starting from material and methods, more in depth analysis about human gait should be performed togheter with simulations  to understand if the mechanism can hurt the patients while supporting, since the cable while supporting can actuate a dangerous tension force on the leg. Please detail which safety procedure have been taken and ho this problem has been faced. Please add this paper as reference for biomechanics simulation an use it to strenghten the section:

DOI: 10.1007/978-3-319-09411-3_62

A kinematic scheme is needed more than figure 2, please add and explain the mechanics actin on the system to undestand how the contro l is performed.

Detail more the control part relating it to kinematics that have to be added.

Results are taken into consideration without using a further system to control the gait. Please take into consideration other paper that did this analysis and implement a comparison of known system to your results. The walking has to be acquired also with another system for validation as a vicon one for example. Please add this paper as reference and use it to strenghten your resul parts that lack on validation with external systems.:

DOI: 10.3390/app10175781

In the discussion a detailed gait analysis as mentioned before should be discussed.

The conclusion is very brief and should be improved.

Appendix A can be put in the text, Appendices are not necessary.

Finally, ethical approvement is missing due to the nature of the experiment please provide the cosent of the subject/s and add the ethical approvement document number and name of the comitee wth date of approval.

Author Response

(The authors gave the same response as above.)

Reviewer 3 Report

The paper proposed an ergonomic design based on the biological structures of lower limbs. The work is quite interesting. The paper is also well-written. However, there are several comments that need to be addressed before considering further publication.

1. What is the difference between the paper and the previous work by the same author, such as in a paper entitled "Ergonomic Design and Performance Evaluation of H-Suit for Human Walking" and ref [39]. Please discuss the previous works in the introduction, so the reader knows about the novelty. 

2. How is the error of the fitted model for Eq (6).

3. Please check equation (6). Is it the fitting equation for force or stiffness?

4. Please add citation or ref for iterative learning control (ILC)

5. Check the fully spelling of the abbreviation ILC on pages 9 and 6. It should only be mentioned once.

Author Response

(The authors gave the same response as above.)

Round 2

Reviewer 2 Report

Auhors improved the paper according to reviewers' suggestion .